# Anaerobic Digestion as an Alternative to Improve the Industrial Production of MnP Economically and Environmentally Using Olive Mill Solid Waste as the Substrate

**DOI:** 10.3390/foods14111918

**Published:** 2025-05-28

**Authors:** Michael Araneda, Fernanda Pinto-Ibieta, Bernabé Alonso-Fariñas, Fernando G. Fermoso, Gustavo Ciudad

**Affiliations:** 1Doctorate in Engineering Sciences with Specialization in Bioprocesses, Universidad de La Frontera, Temuco 4811230, Chile; m.araneda07@ufromail.cl; 2Departamento de Procesos Industriales, Facultad de Ingeniería, Universidad Católica de Temuco, Casilla 15-D, Temuco 4780000, Chile; fpinto@uct.cl; 3Departamento de Ingeniería Química y Ambiental, Escuela Técnica Superior de Ingeniería, Universidad de Sevilla, Camino de los Descubrimientos s/n., 41092 Seville, Spain; bernabeaf@us.es; 4Instituto de la Grasa (C.S.I.C.), Campus Universidad Pablo de Olavide, Edificio 46, Ctra. de Utrera, km. 1, 41013 Sevilla, Spain; fgfermoso@ig.csic.es; 5Departamento de Ingeniería Química, Universidad de La Frontera, Av. Francisco Salazar 01145, Temuco 4811230, Chile; 6Centro de Excelencia en Investigación Biotecnológica Aplicada al Ambiente (CIBAMA), Universidad de La Frontera, Temuco 4811230, Chile; 7Instituto del Medio Ambiente (IMA), Universidad de La Frontera, Avenida Francisco Salazar #01145, Casilla 54-D, Temuco 4780000, Chile

**Keywords:** manganese peroxidase, olive mill solid waste, anaerobic digestion, life cycle assessment, economic assessment, enzyme extract

## Abstract

Manganese peroxidase (MnP) is widely studied for its potential in bioremediation, although its production typically relies on costly synthetic culture media (SCM). This study evaluates olive mill solid waste (OMSW) as a sustainable substrate for MnP production. Three alternatives were evaluated: (1) using SCM; (2) using OMSW; and (3) using OMSW, followed by anaerobic digestion (AD). The alternatives were evaluated by both an economic and life cycle assessment (LCA). The economic analysis considered indicators such as net present value (NPV), internal rate of return (IRR), and payback period. The LCA methodology was conducted according to ISO 14040/44 standards, with a cradle-to-gate system boundary, using SimaPro v9.4 software. Replacing SCM with OMSW improved economic performance, though environmental impacts showed no significant improvement and, in some cases, worsened. In contrast, combining OMSW with anaerobic digestion enhanced both dimensions: Alternative 3 reached the highest NPV (USD 984,464), a 20.9% IRR, and a 4.1-year payback, while reducing impacts by 275% (Stratospheric ozone depletion), 89% (terrestrial ecotoxicity), 78% (freshwater ecotoxicity), and 50% (marine eutrophication) compared to Alternative 1. Finally, the use of OMSW combined with AD reduces economic costs and environmental impact, contributing to the field of sustainable enzyme production

## 1. Introduction

Manganese peroxidase (MnP) is a ligninolytic enzyme produced in most white-rot fungi (WRF) and some bacteria [1]. MnP has drawn immense research attention for its oxidative potential and environmental bioremediation capacity [2,3,4] and has been investigated in several biotechnological applications, such as biopulping in paper production [5,6,7,8,9], discoloration of textile wastewater [10,11,12,13], degradation of antibiotics such as tetracycline and oxytetracycline [14,15], pentachlorophenol degradation [16,17,18], herbicide degradation [19], and remediation of water and soils contaminated with polycyclic aromatic hydrocarbons (PAHs) [3,20,21], among other applications. Culture medium preparation (chemicals addition and sterilization), fermentation of the culture medium by WRF (enzymatic production of MnP), solid–liquid separation carried up by centrifugation and filtration (to separate the liquid phase, rich in MnP, from the solid phase composed of residual WRF and culture medium), and MnP concentration stage by ultrafiltration are the fundamental steps in the production process of liquid solution extract rich in MnP [8,22].

The reduction of MnP production costs using agro-industrial waste as a carbon source has been reported by Lú-Chau et al. [23], who show that MnP production by *Irpex lacteus* using a hemicellulose-rich stream from organosolv pretreatment of beech wood reduces production costs by 1109 times the average price of the product marketed by Sigma-Aldrich (7597 USD/g of MnP). Araneda et al. [2] used olive mill solid waste (OMSW) as a substrate for *Anthracophyllum discolor*, demonstrating that MnP production is the predominant activity compared with laccase and lignin peroxidase. The lignin content of 33.4% and the C/N ratio between 36.5 and 58 in OMSW can act as both a nutrient source and physical support for enhancing WRF growth. In this sense, a C/N ratio of 45 has been reported to result in higher enzymatic activity than ratios 19, 32, and 58 [2]. Polyphenols and hemicellulose in OMSW can also facilitate MnP production and contribute valuable antioxidants [24,25]. Therefore, economic and environmental assessments could yield important insights into the project’s feasibility before industrial implementation. Moreover, this becomes even more relevant in Chile, where, unlike in European countries like Spain, where OMSW is increasingly valorized through various strategies, it remains a waste stream. No large-scale valorization processes are currently implemented, and the waste is commonly sent to accumulation tanks without further treatment or recovery. OMSW is generated from the two-stage production of olive oil; it is estimated that four tons of OMSW can be generated for each ton of olive oil produced [26]. In 2022, Chile produced 16,000 tons of olive oil, where 68% of this production was in the country’s central zone (between the Metropolitan and Maule regions) [27]. Thus, around 43,500 tons of OMSW could be used for MnP production as a substrate to improve the economic profitability of MnP production. However, the economic profitability reported for MnP production using waste does not necessarily indicate environmental benefits [22,23]. Therefore, economic and environmental assessments could yield important insights into the project’s feasibility prior to industrial implementation.

A life cycle assessment (LCA) is a valuable tool for identifying and quantifying the critical processes contributing to the environmental and economic profile [28,29]. To our knowledge, LCA for MnP production has only been reported by González-Rodríguez et al. [22], where wheat straw was used as a carbon source by *Irpex lacteus*, highlighting that energy requirements, particularly electricity and steam consumption, were the main contributors to environmental impacts. Approximately 77, 72, and 85% of the contributions to global warming, fossil resource scarcity, and terrestrial acidification, respectively, were attributed to the high temperature (210 °C) involved in the pretreatment of wheat straw. This underscores the importance of developing alternatives to minimize the energy required for MnP production and reduce environmental impacts. Van den Oever et al. [30] demonstrated that producing 1 MJ of energy using biogas from the anaerobic digestion (AD) from animal manure reduces greenhouse gas emissions by 131% compared to producing the same energy using diesel. In addition, compost production from AD digestate can avoid or decrease the use of synthetic fertilizers, reducing greenhouse gas emissions, environmental impacts associated with fossil resource depletion and freshwater eutrophication [31,32,33]. In this context, whether integrating an AD plant in the MnP process production using OMSW is a viable solution to improve economic and environmental profitability needs to be evaluated.

Therefore, this study aims to perform both an economic and LCA of MnP production by *A. discolor*, a Chilean native WRF selected for its high MnP production capacity. Three production alternatives were evaluated: (1) use of synthetic culture medium (SCM), (2) use of OMSW as the substrate, and (3) use of OMSW as the substrate with an AD stage. The comparison between Alternatives 1 and 2 explores the potential benefits of using agro-industrial waste, while the comparison between Alternatives 2 and 3 assesses the effect of integrating AD. This study is the first to jointly address economic and environmental dimensions of MnP production using *A. discolor* with SCM and OMSW substrates. The results are structured to present investment and operating costs, key profitability indicators, and environmental impacts across nine midpoint categories. Finally, the performance of each alternative is contextualized through a comparative analysis with previously reported enzyme production processes.

## 2. Materials and Methods

### 2.1. Description of the Alternatives for the Production of MnP by A. discolor

Different MnP production alternatives by *A. discolor* were established to evaluate its economic and environmental profile. Alternative 1 uses a SCM as a substrate for producing MnP (proposed as a baseline). In Alternative 2, OMSW is used as a substitute for the SCM. Finally, Alternative 3 contemplates the use of OMSW as a substrate and includes the implementation of an anaerobic digestion (AD) stage of the residual biomass obtained after fermentation, producing by-products such as biogas (used for steam and electricity production for self-consumption) and compost. The MnP production yields using *A. discolor* cultivated in OMSW were experimentally obtained by the authors of the present study and are reported in Araneda et al. [2]. The proposed stages of each alternative of MnP production were selected according to the literature, composed mainly for the preparation of culture medium, fermentation, and liquid–solid phase separation [8,22]. All fermentation alternatives were designed based on an aqueous medium. For each alternative, it was assumed that the MnP production plant was in the central zone of Chile (Maule region), specifically in the olive oil industry, and that the electricity consumed was from the Chilean national grid. Further details regarding the general considerations of the process flow diagram are provided in Appendix A.

#### 2.1.1. Boundaries of the MnP Production for Alternatives 1 and 2

Figure 1a indicates the flows and boundaries of the MnP production using SCM or OMSW as the substrate (Alternatives 1 and 2, respectively) to obtain a liquid solution with >95% of MnP [2]. It is assumed to reach an enzymatic activity of 5000 U/L. The production stages were as follows. (1) Mixer: The culture medium is prepared, which can be either SCM or OMSW. Only in Alternative 1 is the pH adjusted. (2) Sterilization: The culture medium is sterilized with heat produced by a boiler. (3) Fermentation: The sterile culture medium is inoculated with *A. discolor* and cultured for 15 days, producing a final culture and biogenic CO_2_. (4) Centrifugation: The final culture is centrifuged, obtaining enzymatic liquor and residual biomass. (5) Ultrafiltration: The enzymatic liquor is concentrated to obtain MnP, and the generated liquid effluent is treated in a municipal wastewater plant. (6) Thermochemical treatment: The residual biomass is thermochemically treated by applying calcium oxide at 90 °C, obtaining a soil improver for agricultural use [8]. (7) Boiler: Water and diesel produce heat in the boiler, which emits gases into the air and supplies steam for sterilization, fermentation, and thermochemical treatment stages. Finally, the Chilean national grid supplies the electricity to the mixer, centrifugation culture final, and ultrafiltration stages.

#### 2.1.2. Boundaries of the MnP Production for Alternative 3

Figure 1b shows the boundaries of the MnP production by Alternative 3 to obtain a liquid solution with >95% of MnP [2]. It is assumed to reach an enzymatic activity of 5000 U/L. Alternative 3 replaces the thermochemical treatment stage (stage 6 in Figure 1a for an anaerobic digestion (AD) stage (using a continuous stirred tank reactor) to produce biogas and digestate (stage 9, Figure 1b)). The biogas is transferred to a cogeneration engine (10), producing heat and electricity, and the heat is used in the sterilization and AD stages. At the same time, the electricity is used in the mixers (stages 1 and 8), and in the centrifugation final culture, ultrafiltration, AD, and centrifugation digestate stages. Excess heat and electricity leave the system boundaries. The digestate from AD is centrifuged (11) to obtain a dehydrated digestate and liquid effluent stream. A fraction of liquid effluent is sent for a mixer (8) implemented before AD to adjust the concentration of the residual biomass, and the other fraction is discarded as liquid effluent for treatment. Finally, the dehydrated digestate is composted (12), resulting in a compost stream as a by-product and emissions into the air. If the AD does not supply the heat required by the process, a boiler (7) is considered for this.

### 2.2. Anaerobic Digestion (AD) of Residual Biomass from MnP Production Using OMSW

Experimental assays of AD of the residual biomass obtained after fermentation in MnP production were conducted to measure the energy potential and thus evaluate the effect of including an AD stage on the economic and environmental profiles (as described in Alternative 3). The residual biomass for AD was obtained from a previous study that reported the production of MnP with the fermentation of OMSW by *A. discolor* at a concentration of 69 g OMSW/L and a C/N ratio of 58 [2]. At the end of the fermentation stage, a centrifuge was used to obtain a liquid phase rich in MnP, a solid phase of residual biomass (OMSW with WRF). The residual biomass obtained was subjected to AD using the biochemical methane potential (BMP) method by liquid displacement [34]. The characterization of the OMSW and the liquid phase rich in MnP are reported in Appendix A.

The assays were carried out in 250 mL hermetic flasks with a useful volume of 200 mL. BMP was evaluated for 20 days under mesophilic conditions (35 °C). The reactors were mechanically stirred at 300 rpm to facilitate mass transfer. The anaerobic sludge inoculum was collected from the municipal wastewater treatment plant (Temuco, Chile). BMP reactors were loaded with a 2:1 inoculum-to-substrate ratio based on volatile solids (VSs) [34]. The headspace was purged with nitrogen gas to displace oxygen and generate an anaerobic environment [35]. The biogas produced was passed through a 2N NaOH solution to capture CO_2_ and collect the CH_4_ without alteration [36]. All assays were performed in triplicate.

### 2.3. Methodology for the Economic Assessment of MnP Production Using A. discolor

The economic assessment was carried out to determine the production costs of MnP production by Alternatives 1, 2, and 3, respectively. The financial assessment of large-scale MnP production was based on an annual output of 1200 kg of MnP (corresponding to 247 m^3^ of liquid solution with an enzymatic activity of 5000 U/L). To estimate the annual production of MnP, 30% (2100 ton OMSW/year) of the OMSW generated by the Chilean industry Olivares de Quepu was used. Production costs were divided into fixed and variable costs. Fixed costs were based on the personnel costs required for this type of installation (259,416 USD/year) [37]. At the same time, variable costs are those that vary according to the amount of production, such as raw material costs. The input costs for estimating variable costs (unit prices) and the parameters for economic assessment are shown in Table 1. For Alternatives 2 and 3, it is assumed that OMSW is considered waste in the olive oil industry and, therefore, has no economic value as a raw material for MnP production. All costs were calculated based on Chilean market conditions and literature reports [38,39,40,41,42]. The currency used was the United States dollar (USD), and the conversion rate from the Chilean peso (CLP) to USD was CLP 889.21 [43]. A lifetime of 25 years and an operating time of 330 days/year were used. A discount rate of 13.6% was used [37]. The taxation rate was 35%, as reported by the World Bank for Chile [44]. The 10-year straight-line method was used for depreciation [37].

Total capital investment (TCI) was considered without a mortgage and considered only the main equipment of each MnP production alternative, as well as the storage tanks and piping. The main process units were sized for two monthly cycles in discontinuous production. The sizing of the required equipment was separated into a MnP production plant and an energy production plant. The costs of the productive units were estimated according to local suppliers in Chile and scaling criteria reported by Coral-Velasco et al. [37]. Piping and construction costs were assumed to be 14% of the value of the main equipment [37]. The investment cost of the energy production plant (AD and cogeneration) was estimated using the factor USD 3180 per installed kWe for the construction of the anaerobic digester and the cogeneration biogas engine facilities [25].

Net present value (NPV), internal rate of return (IRR), and payback period (PBP) indicators were evaluated over a 10-year horizon. A minimum selling price (MSP) was estimated for the three production alternatives where the NPV was zero; this element is critical to determining profitability [50]. Final indicators for Alternatives 1 and 2 were estimated with an assumed selling price (ASP) of 10% above the MSP. Meanwhile, in Alternative 3, the same ASP as in Alternative 2 was used.

### 2.4. Environmental Assessment

The environmental assessment for the three MnP production alternatives was carried out using the life cycle assessment (LCA) methodology according to ISO 14040/44 standards [51,52].

#### 2.4.1. Goals and Scope of the Life Cycle Assessment (LCA)

The LCA was performed to estimate and compare the environmental profiles of three MnP production alternatives described in Section 2.1. The system boundary of the study is cradle to gate. The function of the systems under study is to obtain a MnP-rich liquid solution with an enzymatic activity of 5000 U/L for use in the degradation processes of persistent contaminants or other biotechnological applications. The functional unit was 1 kg of MnP production with the mentioned enzymatic activity. The MnP production was considered to be a finished product at the process gate. Therefore, the environmental impacts of transportation, marketing, and use are not included. All liquid effluents generated were treated in a wastewater plant for final disposal until the end of life.

For co-products like compost, soil improver, and electricity, system expansion was applied as recommended by ISO 14040/44 standards [51,52]. Compost and soil improvers were credited with avoiding the production of ammonium nitrate, as it is the most widely used fertilizer in olive fields in Chile [53]. Therefore, avoiding its production implies credits associated with the environmental impacts derived from its use. Additionally, electricity was credited with avoiding production of medium-voltage Chilean mixed. More details are in Appendix A.

Assumptions and limitations are reported in Appendix A.

#### 2.4.2. Life Cycle Assessment (LCA) Inventory Data

The previous literature and research provided the magnitudes of the input and output requirements for conducting the inventory data (data available in Appendix A). The Ecoinvent V.3.8 database was used for background inventory data.

#### 2.4.3. Environmental Impact Assessment

The SimaPro v.9.4 software from PRé Sustainability B.V. (Amersfoort, The Netherlands) was used to model the LCA. The environmental impact was calculated using the ReCiPe 2016 hierarchist midpoint method V1.07 World (2010) [54], which includes several impact indicators, from which the following were selected according to the literature [22]: global warming (GW), stratospheric ozone depletion (SOD), terrestrial acidification (TA), freshwater eutrophication (FE), marine eutrophication (ME), Terrestrial ecotoxicity (TET), Freshwater ecotoxicity (FET), Marine ecotoxicity (MET) and Fossil resource scarcity (FRS).

## 3. Results and Discussion

### 3.1. Economic Assessment for MnP Production by A. discolor

#### 3.1.1. Equipment Size and Investment Costs for the Different MnP Production Alternatives

Table 2 details the equipment size and the estimation of the initial investment required to produce 1200 kg of MnP annually. Alternatives 2 and 3 require 150% more culture medium, with 1000 m^3^/month compared to 400 m^3^/month for Alternative 1. This increase is driven by the lower MnP productivity using OMSW as the substrate, which achieves 0.1 kg MnP/m^3^ compared to 0.25 kg MnP/m^3^ when the SCM is used [2,55]. Consequently, to achieve MnP productivities similar to SCM, Alternatives 2 and 3 required an increase in the equipment sizes, as larger volumes of culture must be processed, which, combined with the higher density of OMSW (69.3 g/L) compared to SCM (14.6 g/L), leads to increased sterilization energy demands for OMSW-based culture medium.

Alternative 2 has the highest fuel storage and boiler requirements to cover external fuel demand for steam production. Nevertheless, in Alternative 3, requirements for fuel storage and boiler sizes remain smaller (9 m^3^ and 13 ton/h, respectively) despite increased culture medium volumes. This result in Alternative 3 is achieved by covering the part of fuel needs for steam production with the biogas generated in the AD stage. As illustrated in Appendix A, the calculated methane production from the AD of the residual biomass after fermentation in the MnP production process using OMSW reaches 171 ± 2 mL CH_4_/g VS, which aligns with the average reported in the literature for biogas production from OMSW, ranging from 34 to 345 mL CH_4_/g vs. [56]. Considering a heating value of methane of 36,000 KJ/m^3^ [57], the energy equivalent is 6.2 MJ/kg vs. of residual biomass obtained after the fermentation stage. This energy is utilized to produce heat and electricity, reducing steam requirements by 38% compared to Alternative 1. The efficient use of biogas minimizes reliance on external energy sources. It underscores the strategic advantage of Alternative 3 in optimizing resource utilization and reducing infrastructure expansion related to external energy production.

The initial investment costs reflect the differences in equipment requirements among the alternatives, amounting to USD 1,512,441; 2,832,464; and 3,231,155 for Alternatives 1, 2, and 3, respectively. Although Alternative 3 reduces the size of the equipment associated with external energy, it has a higher investment cost due to its implementation of the AD and cogeneration plant.

#### 3.1.2. Economic Assessment of Different MnP Production Alternatives

Table 3 presents the economic assessment for each alternative MnP production, showing a significant decrease in variable costs in Alternative 3 compared to the other alternatives. OMSW reduces variable costs by 81%, and incorporating anaerobic digestion (AD) further cuts variable costs by 93%, making Alternative 3 the most economical option in terms of costs. In Alternative 1, the highest costs come from glucose and peptone, which account for 80% of all inputs (Table 1). These findings align with Lú-Chau et al.’s [23] report on MnP production by *Irpex lacteus* using synthetic media, where approximately 50% of variable costs were attributed to chemical reagents, particularly peptone, which alone contributed 30% of variable costs (around USD 165 of peptone per kg of MnP). The variable costs in Alternatives 2 and 3 were 96 and 99% lower, respectively, than the 7597 USD/g of MnP average price of the product marketed by Sigma-Aldrich reported by Lú-Chau et al. [23].

Alternative 1 requires a minimum sales price (MSP) of 2083 USD/kg to make no profit, and this value drops to 1060 and 931 USD/kg for Alternatives 2 and 3, respectively. Conversely, with an assumed sales price (ASP) of 2291 USD/kg in Alternative 1, it decreases by 49% (1166 USD/kgMnP) in the alternatives that employ OMSW. For the net present value (NPV) and profit margin, the highest NPVs of USD 984,464 and profit margin of USD 1,016,280 were achieved in Alternative 3. This advantage is due to the significantly lower variable costs in Alternative 3 that offset the higher initial investment needed for the AD and cogeneration plant. In comparison, while Alternative 2 also utilizes OMSW, it has a lower NPV of USD 442,987 and a profit margin of USD 780,866, which is insufficient to fully cover its investment costs. Alternative 1, with an NPV of USD 873,444 and a profit margin of USD 602,299, remains less favorable than Alternative 3 regarding overall profitability. Although Alternative 1 has the highest IRR (27.1%) and the shortest payback period (3.4 years), Alternative 3 remains competitive with a 20.9% IRR and a 4.1-year payback period. Alternative 2, with an IRR of 17.5% and a payback period of 4.6 years, is the least favorable in terms of profitability and investment recovery.

Due to the paucity of economic studies on MnP production, a comparison was made with the production of the cellulase enzyme using genetically modified *Trichoderma reesei* on coffee husk as a substrate, as reported by Coral-Velasco et al. [37]. The results achieved a payback period of 2.3 years and an IRR of 61%, based on an annual sales volume of 724,000 kg of cellulase at a market price of 42 USD/kg. In contrast, our analysis shows a 600-times-lower sales volume and a 28-times-higher sales price. This economic differential may stem from the high demand for cellulase and its production advances that have reduced variable costs, positioning it as the third best-selling enzyme on an industrial scale [59], with extensive applications such as biofuels, textiles, food, and paper [60], thereby enabling its large-scale production. At the same time, MnP, although relevant in bioremediation, faces a considerably lower demand. However, the IRR obtained for the three MnP production alternatives analyzed in our study remains competitive, exceeding the average IRR of 10.7% reported for the global chemical industry [37].

In conclusion, the assessment of the three MnP production alternatives shows that Alternative 3 has the highest NPV, and its IRR and payback are competitive, making it the most profitable option. Therefore, using OMSW as a substrate and implementing an AD stage in combination yields benefits such as reduced variable costs and better annual profit margins than SCM. Therefore, adding an AD stage could significantly improve the economic indicators despite the higher investment.

### 3.2. Life Cycle Assessment (LCA) for MnP Production by A. discolor

Figure 2 shows the percentage contribution to the nine impact categories evaluated for the different MnP production alternatives. For each MnP production alternative, water, CaCl_2_, and MgSO_4_ showed a lower-than-6% contribution in all environmental impact categories evaluated. In contrast, the effects of glucose, KH_2_PO_4_, MnP production plant, diesel, liquid effluent, and electricity contributed over 30% in at least one impact category.

Glucose contributed only in Alternative 1, with contributions ranging from 57 to 69% observed in ME, SOD, and FET. Similarly, the KH_2_PO_4_ contributed only in Alternative 1, mainly in the TET (48%) and MET (33%) categories. Due to the use of SCM, glucose and KH_2_PO_4_ contributed to the impact categories associated only with Alternative 1, while their replacement by OMSW in Alternatives 2 and 3 reduced the impacts associated with using these chemicals for the culture-medium preparation. The MnP production plant affects the GW and TA impact categories across all the studied alternatives. Depending on the alternatives analyzed, the contribution ranges from 66 to 81% for GW and from 32 to 59% for TA. The contribution to GW and TA involved within the MnP production plant is mainly due to the air emissions by burning diesel in the boilers to produce heat for the thermal treatment stage in Alternatives 1 and 2 and the sterilization stage in all alternatives (Figure 1, Appendix A, and Table 4).

Furthermore, diesel, as the environmental impact associated with its production, affects all impact categories except for ME in all alternatives studied, incrementing its contribution in Alternative 2 and more in Alternative 3. Thus, for Alternative 1, FRS and FE show contributions between 60 and 80%. In contrast, in Alternative 2, contributions exceed 80% for FRS, TET, and FE, and Alternative 3 indicates FRS with contributions between 80 and 100%, while TET and FE range between 50 and 60%. The impacts associated with diesel burning and production primarily arise from the energy needs for the sterilization stage, considering that this stage accounts for more than 88% of the total steam requirements in all the alternatives (Table 4). Consistent with, LCA performed for enzyme production showed that the significant contributions to environmental impacts were associated with the process that requires energy input (heat and electricity), such as the agitation of fermentation and sterilization [61] or pretreatment, when waste is used (wheat straw) as the substrate [22]. Liquid effluent to treatment contributed to MET, FE, and FET between 27% and 35% for Alternative 3; nevertheless, it significantly influenced ME in Alternatives 2 and 3, contributing over 70%. Nevertheless, credits per product avoided ammonium nitrate only showed a positive effect on the SOD impact category by −17 to −56% across all alternatives. Conversely, credits-per-product avoided electricity positively influenced all impact categories studied only for Alternative 3, contributing in a range from −25 to −52% for SOD, TET, and FET.

Figure 3 compares (in percentages) the environmental impact values generated per each MnP production alternative evaluated, considering Alternative 1 as the baseline (Table 5). Compared with Alternative 1, the MnP production through Alternative 2 increased the impact value for FRS, FE, and GW by 150 to 160%. The SOD decreased by 128%, and ME, TET, and FET varied between 21 and 32%. Alternative 3 shows a decrease in the value of all impact categories concerning Alternative 1. A reduction of 275% was observed in SOD, while for the remaining categories, the reduction was 17 to 89%.

According to Table 5, GW and TA impacts are higher in Alternative 2 than in Alternative 1, with values of 879 versus 351.3 kg CO_2_ eq for GW and 2.5 versus 1.3 kg SO_2_ eq for TA. This increase is mainly due to the lower MnP yield obtained when using OMSW as a substrate, which requires processing larger volumes of culture medium and greater energy input for sterilization. Although MnP yields in Alternative 3 are equal to those in Alternative 2, GW and TA impacts are reduced by 69 and 64%, respectively. These reductions are attributed to the heat generated from biogas in the AD stage (i.e., 2033 MJ/kg MnP produced). As a result, carbon emissions decrease from 879 to 271.5 kg CO2 eq/kg MnP, and sulfur dioxide emissions from 2.5 to 0.9 kg SO_2_ eq/kg MnP.

Nevertheless, heat production from the AD stage in Alternative 3 covers around 53% of the total system heat internal requirements, so diesel consumption is still required (Appendix A). Thus, the results show different diesel consumption, 75, 233, and 75 kg of diesel/kg MnP, for Alternatives 1, 2, and 3 (Appendix A), strongly impacting FRS, which, in Alternative 2, is higher by 210% than in Alternative 1, and in Alternative 3 is the same as Alternative 1. In this vein, the electricity (340 kWh/kg MnP) produced in the cogeneration implemented in Alternative 3 is enough to cover all system demands and generate surpluses of 158 kWh/kgMnP produced (see Appendix A), enabling reductions in FE, TET, FET, and MET in Alternative 3. The SOD category decreases both Alternatives 2 and 3 by 128% and 275% compared to Alternative 1 due to the replacement of SCM by OMSW, no glucose being required, and product credits for ammonium nitrate being avoided. Nevertheless, SOD reduction in Alternative 3 is 528% lower than in Alternative 2 due to the AD stage implemented to procure electricity, which increases avoided credits for electricity by 52%.

In summary, the AD implementation enables the supply of electricity and heat, and a digestate that can be used as fertilizer. The electricity production reduces the impact in all categories, while the heat production mainly contributes to the reduction of impact on GW and TA (primarily associated with diesel burning); the decrease in FE, TET, FET, MET, and FRS (mainly associated with diesel production); and the compost production after AD reduces the SOD category. Thus, the application of AD is crucial to making the production of MnP using OMSW as a substrate environmentally viable.

Compared to LCA for the MnP production using wheat straw performed by González-Rodríguez et al. [22], the midpoint values were 2852 kg CO_2_ eq, 11.2 kg SO_2_ eq, and 864 kg oil eq for GW, TA, and FRS, respectively (Table 5), ten times more than reported for Alternative 3 proposed in our study. According to González-Rodríguez et al. [22], about 77, 72, and 85% of the contributions for GW, FRS, and TA are due to the pretreatment of wheat straw, which requires 3019 MT/batch of steam demand due to the 210 °C the process requires. Therefore, this indicates that generating alternatives to minimize the energy needed for the sterilization stage is critical to reducing the environmental costs associated with the MnP production process. In this context, using OMSW is crucial for better environmental performance in MnP production, as it does not require pretreatment, thereby avoiding the associated environmental impacts. Moreover, it has been reported that enhancing environmental performance increases enzyme production yields, which could lead to a lower environmental impact [9]. Hence, optimizing MnP production yields using OMSW to reduce the required volume for sterilization could result in a more sustainable process.

Alternative 3 demonstrates reductions of about a thousand times less impact for GW, TA, TET, MET, and FRS, compared to the midpoint values obtained in the LCA for cellulase production from coffee husks, as reported by Catalán et al. [58], where the values were 425,722 kg CO_2_ eq, 0.2 kg CFC-11 eq, and 100,436 kg oil eq for GW, SOD, and FRS, respectively (Table 5). In total, 94% of the GW, FRS, and TET impacts reported by Catalán et al. [58] were attributed to the lyophilization stage, which is not included in the LCA analysis for MnP production using OMSW. Therefore, to estimate the contributions from all process steps except lyophilization, 6% of these midpoint values were calculated (25,543 kg CO_2_ eq, 6026 kg oil eq, and 14,654 kg 1,4-DCB eq for GW, FS, and TET). The results obtained remain higher than that reported by Alternative 3. Even when lyophilization was excluded, the impacts were primarily attributed to energy demand. Still, they were associated with aeration compressors and temperature control during fermentation. So, even when excluding the effects of the major contribution stage in cellulase production (lyophilization), MnP production demonstrates a significant environmental advantage, with reductions of approximately 99.9% for GW, FRS, and TET mainly due to the energy obtention from AD-stage implementation. This underscores the need to replace the energy source with less polluting alternatives, such as AD.

### 3.3. Final Remark and Future Directions

Alternative 3 offers substantial economic and environmental advantages over Alternatives 1 and 2. Compared with other studies, Alternative 3 demonstrates economic competitiveness in terms of MnP production costs using a side stream of the organosolv process [23]. However, compared with a high-demand industrial enzyme like cellulase, the economic indicators for MnP production in Alternative 3 are lower. Nonetheless, they remain profitable by chemical industry standards [37] and can further improve once MnP production scales to an industrial level. Despite modest current economic indicators for MnP production using OMSW, the environmental impacts associated with MnP production through the process proposed in Alternative 3 are promising, showing impacts up to ten times lower than those reported by González-Rodríguez et al. [22] for MnP production using wheat straw as a substrate, and even a thousand times lower than those documented by Catalán et al. [58] for cellulase production using coffee husk. In this sense, the lower production costs of the cellulase enzyme do not imply better benefits in environmental terms than Alternative 3 of our study. These results demonstrate the importance of evaluating the economic and environmental aspects of MnP production for industrial-scale production. The significant improvements in both economic and environmental aspects observed in Alternative 3 over Alternatives 1 and 2, as well as the reduced environmental impacts relative to MnP production from wheat straw and cellulase from coffee husks, can be primarily attributed to the integration of the AD process and the use of OMSW as a substrate.

The critical advantage of AD implementation in Alternative 3 is its ability to significantly reduce diesel consumption through the cogeneration of electricity and steam using biogas derived from AD. This system fully meets the electricity requirements and reduces steam demand, which is mainly required for the sterilization stage. Although implementing a cogeneration plant involves a higher initial investment, the long-term savings from reduced diesel and electricity consumption make Alternative 3 more economically viable than Alternatives 1 and 2, which lack AD and face higher energy costs. According to Campello et al. [62], biogas can produce significant amounts of electricity and heat, reducing reliance on external energy sources and lowering operational costs. Additionally, as Gaffey et al. [63] highlight, the electricity generated could be sold, offering a significant financial return in countries with grid-access options. The reduction in diesel consumption, resulting from using biogas as an energy source, leads to a substantial decrease in CO_2_ and SO_2_ emissions in Alternative 3. In contrast, Alternatives 1 and 2 depend entirely on fossil fuels for steam generation, contributing to increased greenhouse gas emissions.

In addition, using OMSW as the substrate in Alternatives 2 and 3 reduces production costs compared to using SCM in Alternative 1. However, the AD in Alternative 3 maximizes these savings by further reducing the production costs associated with energy use. In contrast, Alternative 2, without AD, does not benefit from the same level of cost reduction. Although Alternative 3 involves larger equipment and higher capital costs, the long-term economic benefits make it more cost-effective. Mendieta et al. [32] support the environmental benefits of AD by demonstrating its capacity to mitigate freshwater eutrophication through the use of digestate as a biofertilizer, which reduces the need for synthetic fertilizers. In addition, fertilizers derived from AD can improve the profitability of the process by creating an additional source of revenue, which increases the overall return on project investment [31,33]. This study contributes to the field by introducing an innovative biotechnological strategy that integrates OMSW and AD for the sustainable and economically viable production of MnP. To the best of our knowledge, this is the first study that comprehensively evaluates MnP production from OMSW using *A. discolor*, a Chilean native WRF, while simultaneously assessing environmental and economic performance through LCA and techno-economic indicators. The innovation lies in coupling enzymatic production with AD, a strategy that not only valorizes agro-industrial residues but also mitigates the environmental burden typically associated with enzyme production using synthetic culture media. This integrated system enables biogas and compost generation, reducing fossil energy dependency and supporting circular-economy principles. In summary, this research presents a dual-benefit strategy that recovers value from waste while establishing a feasible and scalable route for MnP production with significantly reduced environmental impacts and favorable economic returns.

## 4. Conclusions

While using OMSW in Alternative 2 provides cost reductions and waste valorization, it introduces significant environmental and operational challenges, mainly due to high diesel consumption and low MnP yields. AD implementation in Alternative 3 addresses these issues by reducing diesel use, generating surplus electricity, and adding compost production, thus reducing the environmental impacts in all evaluated categories. In addition, it enhances economic performance through cost savings and additional revenue streams. Combining OMSW with AD is crucial for achieving a balanced, sustainable, and economically viable MnP production process using *A. discolor*. Thus, this research suggests that using OMSW as a substrate for MnP production is a sustainable and feasible way to implement it on an industrial scale. These results contribute to the sustainable production of enzymes and the valorization and management of organic waste. This is the first study to explore MnP production from OMSW using *A. discolor*, while also incorporating environmental and economic aspects, often overlooked in studies focused mainly on enzymatic yields.

## Figures and Tables

**Figure 1 foods-14-01918-f001:**
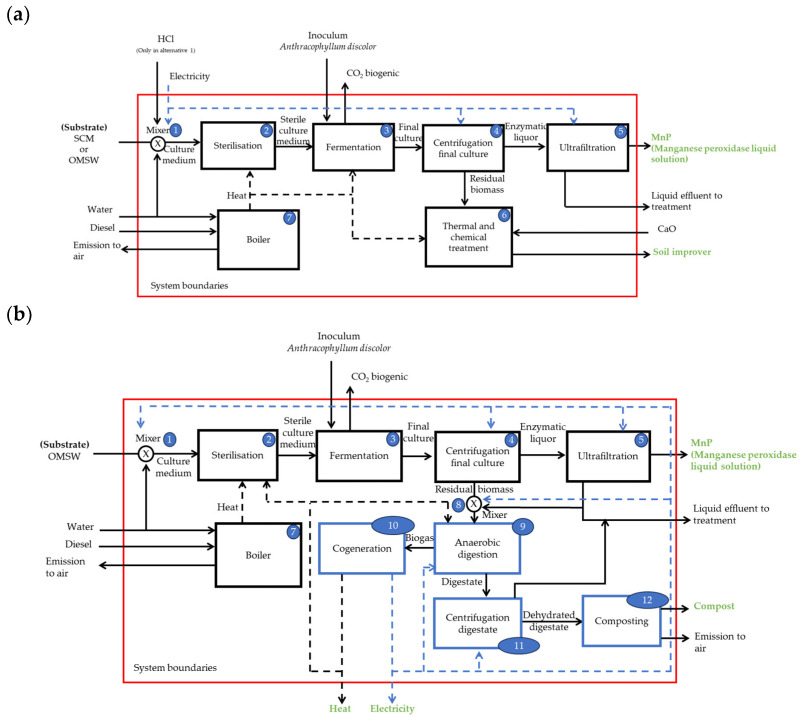
MnP production boundaries for Alternatives 1 and 2 (**a**). MnP production boundaries for Alternative 3 (**b**). The lines describe the flow directions; the black dotted lines describe heat flows, and the blue lines describe electricity flows.

**Figure 2 foods-14-01918-f002:**
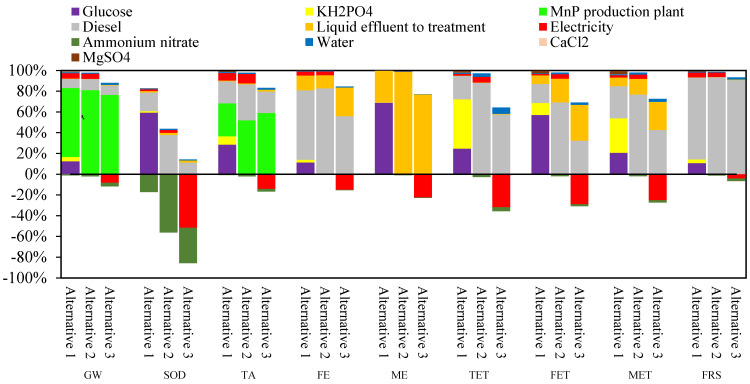
Percentage contribution of environmental impacts for manganese peroxidase production by all alternatives (GW, global warming; SOD, stratospheric ozone depletion; TA, terrestrial acidification; FE, freshwater eutrophication; ME, marine eutrophication; TET, terrestrial ecotoxicity; FET, freshwater ecotoxicity; MET, marine ecotoxicity; FRS, fossil resource scarcity).

**Figure 3 foods-14-01918-f003:**
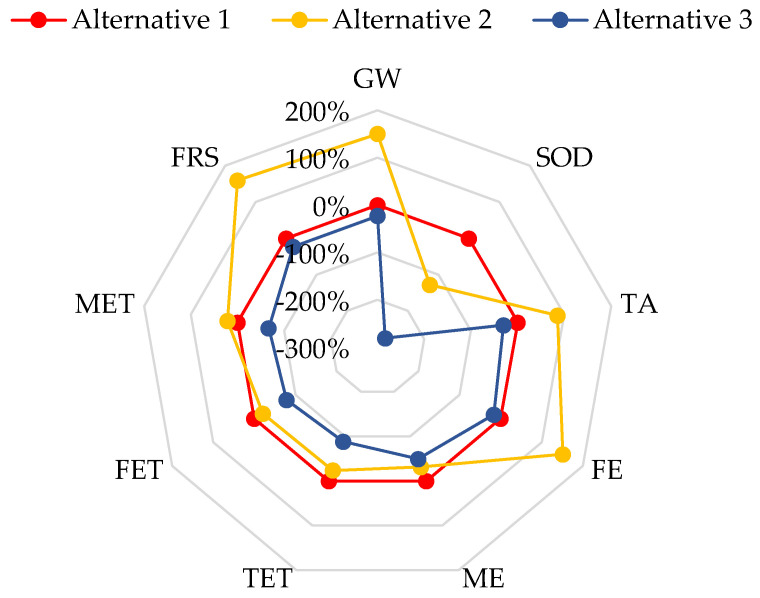
Comparison of the estimated environmental impacts for each MnP production alternative (GW, global warming; SOD, stratospheric ozone depletion; TA, terrestrial acidification; FE, freshwater eutrophication; ME, marine eutrophication; TET, terrestrial ecotoxicity; FET, freshwater ecotoxicity; MET, marine ecotoxicity; FRS, fossil resource scarcity).

**Table 1 foods-14-01918-t001:** Unitary prices for cost estimation and economic assessment parameters.

Item	Price	Unit	Reference
Electricity	0.1	USD/kWh	[40]
Diesel	1.2	USD/kg	[45]
Water	1.7 ^1^	USD/m^3^	[38]
Glucose	17.9	USD/kg	Sigma-Aldrich, St Louis, MO, USA (G8270-25KG)
Peptone	68.6	USD/kg	[42]
KH_2_PO_4_	20.5	USD/kg	[46]
MgSO_4_	7.7	USD/kg	[47]
CaCl_2_	3.5	USD/kg	[41]
Thiamine	95.2	USD/kg	[39]
HCl	14.4	USD/L	[48]
CaO	2.6	USD/kg	[49]
Parameters	Value/Description		
Currency	US dollar		
Conversion factor	USD = CLP 889.21		[43]
Year of analysis	2023		
Production	1200 kg of MnP/year		
Project lifetime	25 years		
Annual operating time	330 days		
Discount rate	13.6%		[37]
Taxation rate (year 2019)	35%		[44]
Depreciation	Straight line method over 10 years		[37]

^1^ Includes wastewater treatment costs; USD, United States dollar; CLP, Chilean peso; MnP, manganese peroxidase.

**Table 2 foods-14-01918-t002:** Equipment size and investment costs for the different MnP production alternatives.

	Equipment Size			Cost of Scaling for Each Equipment (USD)			Reference
	Alternative			Alternative			
	1	2	3	1	2	3	
**Production plant MnP**							
Tank for culture medium (m^3^)	400	1	1	318,215	587,949	587,949	[37]
Pumping flow (m^3^/h)	25	62.5	62.5	10	10	10	LS
Sterilizer (m^3^/h)	25	62.5	62.5	103,047	154,215	154,215	[37]
Fermenter (m^3^)	200	500	500	355,036	484,811	484,811	[37]
Centrifugal decanter (m^3^/h)	25	62.5	62.5	130,009	251,472	251,472	[37]
Ultrafiltration unit (m^3^/h)	25	62.5	62.5	307,224	675,590	675,590	[37]
Seed fermenter (m^3^)	5	5	5	40,571	40,571	40,571	[37]
Tank for biomass (m^3^)	20	277	277	26,989	197,992	197,992	[37]
Tank for fuel (m^3^)	9	27	9	15,612	42,016	15,640	[37]
Boiler (ton/h)	21	147	13	20	40	20	LS
Piping and electrical projects				185,738	347,846	341,353	[37]
**Production plant energy**							
Anaerobic reactor and cogeneration engine (kWe)			142			451,560	[25]
Total investment (USD)				1,512,441	2,832,464	3,231,155	

LS: local supplier in Chile.

**Table 3 foods-14-01918-t003:** Economic assessment of different alternatives for MnP production and comparison with cellulase production.

Item	Alternative 1	Alternative 2	Alternative 3	MnP Production Using SCM and Side-Stream of the Organosolv Process [23]	Cellulase Using Coffee Husk as Substrate [58]
Variable costs (USD/kg MnP)	1573	299	103	n.r.	28.7
Minimum selling price (MSP) (USD/kg MnP)	2083	1.06	931	n.r.	n.r.
Assumed selling price (ASP) (USD/kg MnP)	2291	1166	1166	7597	42
NPV (USD)	873,444	442,987	984,464	n.r.	32,958
IRR (%)	27.1	17.5	20.9	n.r.	61.02
Payback period (year)	3.4	4.6	4.1	n.r.	2.27
Annual profit margin (APM) (USD/year)	602,299	780,866	1,016,280	n.r.	11,956

MnP, manganese peroxidase liquid solution; NPV, net present value; IRR, internal rate of return; APM, (annual sales revenue—annual production costs); n.r., not reported.

**Table 4 foods-14-01918-t004:** Electricity and heat requirements for each stage and MnP production alternative.

Energy	Stage	Alternative 1	Alternative 2	Alternative 3
Electricity (kWh)	Mixer (1)	15.2	38.0	38.0
	Centrifugation final culture (4)	12.0	30.0	30.0
	Ultrafiltration (5)	5.8	12.7	12.7
	Mixer (8)			38.0
	Anaerobic digestion (9)			38.0
	Centrifugation digestate (11)			25.3
**Total electricity**		**33.0**	**80.7**	**182.0**
Heat (MJ)	Sterilization (steam) (2)	1625 (92%)	4219 (88%)	4219 (97%)
	Fermentation (3)	84.6		
	Thermal treatment residual biomass (6)	50.8	593.9	
	Anaerobic digestion (9)			133.4
**Total heat**		**1760 (100%)**	**4813 (100%)**	**4352 (100%)**

**Table 5 foods-14-01918-t005:** Midpoint values per impact category for different MnP production alternatives and comparison with cellulase production.

Impact Category	Unit	Alternative 1	Alternative 2	Alternative 3	MnP Production Using SCM and Wheat Straw as Substrate [22]	Cellulase Production Using Coffee Husk as Substrate [58]
GW	kg CO_2_ eq	351.3	879.0	271.5	2852	425,722
SOD	Kg CFC-11 eq	2.4 × 10^−4^	−6.6 × 10^−5^	−4.2 × 10^−4^	2.0 × 10^−3^	1.9 × 10^−1^
TA	kg SO_2_ eq	1.3	2.5	0.9	11.2	2.6
FE	kg P eq	0.05	0.12	0.04	0.74	16.00
ME	kg N eq	0.07	0.05	0.04	0.39	1.44
TET	kg 1.4-DCB	487	370	56	2.28	244,249
FET	kg 1.4-DCB	0.43	0.34	0.09	22.1	63.9
MET	kg 1.4-DCB	0.58	0.71	0.19	31.13	228.00
FRS	kg oil eq	107	278	82	864	100,436

GW, global warming; SOD, stratospheric ozone depletion; TA, terrestrial acidification; FE, freshwater eutrophication; ME, marine eutrophication; TET, terrestrial ecotoxicity; FET, freshwater ecotoxicity; MET, marine ecotoxicity; FRS, fossil resource scarcity; CFCs, chloro fluoro carbonates; DCB, dichlorobenzene.

## Data Availability

The original contributions presented in this study are included in the article/Appendix A. Further inquiries can be directed to the corresponding author.

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
