# Peer review of "Anaerobic Digestion as an Alternative to Improve the Industrial Production of MnP Economically and Environmentally Using Olive Mill Solid Waste as the Substrate"

_foods, 2025, doi:10.3390/foods14111918_

Round 1
Reviewer 1 Report
Comments and Suggestions for Authors
Minor revision.
It is suggested that adding some descriptions or calculations about the carbon dioxide (equivalent) emission reduction of the whole process under different situations.
It is suggested that describe the replicates and method of statistical analysis in more details so that the contents will be much better.
Author Response
Reviewer #1 comment 1: The abstract should be reformulated and contain the methodology/method and the main results obtained. It should also include the contribution to the field.
Author's response: The abstract has been reformulated and the methodology/method and the main results obtained were incorporated in the following form: “Manganese peroxidase (MnP) is widely studied for its potential in bioremediation, although its production typically relies on costly synthetic culture media (SCM). This study evaluates olive mill solid waste (OMSW) as a sustainable substrate for MnP production. Three alternatives were evaluated: (1) using SCM, (2) using OMSW, and (3) using OMSW followed by anaerobic digestion (AD). The alternatives were evaluated by both an economic and life cycle assessment (LCA). The economic analysis considered indicators such as net present value (NPV), internal rate of return (IRR), and payback period. The LCA methodology was conducted according to ISO 14040/44 standards, with a cradle-to-gate system boundary, using SimaPro v9.4 software. Replacing SCM with OMSW improved economic performance, though environmental impacts showed no significant improvement and, in some cases, worsened. In contrast, combining OMSW with anaerobic digestion enhanced both dimensions: Alternative 3 reached the highest NPV (USD 984,464), a 20.9% IRR, and a 4.1-year payback, while reducing impacts by 275% (Stratospheric ozone depletion), 89% (terrestrial ecotoxicity), 78% (freshwater ecotoxicity), and 50% (marine eutrophication) compared to Alternative 1. Finally, the use of OMSW combined with AD reduces economic costs and environmental impact, contributing to the field of sustainable enzyme production” (lines 38 - 52).
Reviewer #1 comment 2: In the introduction section, there should be a paragraph that presents the structure of the paper.
Author's response: The structure paper was redacted in the introduction section as follows: “Therefore, this study aims to perform both an economic and LCA of MnP production by A. discolor, a Chilean native WRF selected for its high MnP production capacity. Three production alternatives were evaluated: (1) use of synthetic culture medium (SCM), (2) use of OMSW as the substrate, and (3) use of OMSW as the substrate with an AD stage. The comparison between Alternatives (1) and (2) explores the potential benefits of using agro-industrial waste, while the comparison between Alternatives (2) and (3) assesses the effect of integrating AD. This study is the first to jointly address economic and environmental dimensions of MnP production using A. discolor with SCM and OMSW substrates. The results are structured to present investment and operating costs, key profitability indicators, and environmental impacts across nine midpoint categories. Finally, the performance of each alternative is contextualised through a comparative analysis with previously reported enzyme production processes” (lines 116 - 132).
Reviewer #1 comment 3: Figure 1(a) and (b) should present the flow in detail.
Author's response: Additional details on the general considerations underlying the process described in Figure 1(a) and (b) are provided in Supplementary Material S.1, excluded from the main text to preserve clarity due to its level of technical detail (e.g., concentrations, yields, chemical compositions, system boundaries, etc). This additional information is referenced in the Materials and Methods section as follows: "Further details regarding the general considerations of the process flow diagram are provided in Supplementary Material S.1" (lines 151 – 152).
Reviewer #1 comment 4: Add a Discussion section that discusses the innovative elements brought by the paper, what gaps it fills and what are the advantages of the research.
Author's response: The suggestion was incorporated as follows “This study contributes to the field by introducing an innovative biotechnological strategy that integrates OMSW and AD for the sustainable and economically viable production of MnP. To the best of our knowledge, this is the first study that comprehensively evaluates MnP production from OMSW using A. discolor, a Chilean native WRF, while simultaneously assessing environmental and economic performance through LCA and tech-no-economic indicators. The innovation lies in coupling enzymatic production with AD, a strategy that not only valorizes agro-industrial residues but also mitigates the environ-mental burden typically associated with enzyme production using synthetic culture media. This integrated system enables biogas and compost generation, reducing fossil energy dependency and supporting circular economy principles. In summary, this research presents a dual-benefit strategy that recovers value from waste while establishing a feasible and scalable route for MnP production with significantly reduced environmental impacts and favorable economic returns” (Lines 625 – 637).

Reviewer 2 Report
Comments and Suggestions for Authors
Dear Authors,
The paper is interesting and fits within the journal's objectives. However, it requires several improvements.
- The abstract should be reformulated and contain the methodology/method and the main results obtained. It should also include the contribution to the field.
- In the introduction section, there should be a paragraph that presents the structure of the paper.
- Figure 1(a) and (b) should present the flow in detail.
- Add a Discussion section that discusses the innovative elements brought by the paper, what gaps it fills and what are the advantages of the research.
Author Response
Reviewer #2
This article presents an economic analysis of the production of MnP from the fermentation of olive mill solid residue (OMSW) and subsequent anaerobic digestion of the fermented waste. Some corrections are necessary to improve the text:
Reviewer #2 comment 1: The title needs to be changed, as it gives the impression that fermentation processes were carried out, but this did not occur. This is an article of economic analysis only. It is also unnecessary to include the country Chile in the title.
Author's response: To clarify that the MnP production data were obtained experimentally in a previous study by the authors and not within the scope of the present manuscript, the following sentence has been added: “The MnP production yields using A. discolor cultivated in OMSW were experimentally obtained by the authors of the present study and are reported in Araneda et al. [2]” (Lines 142 – 144). In addition, the country "Chile" has been removed from the title.
Reviewer #2 comment 2: Lines 68-69 “This approach can, therefore, transform agroindustrial waste into a valuable biotechnological resource, reducing the environmental impact and supporting” What is the conventional destination of OMSW? If there is a destination, even if it is less noble, then it cannot be considered waste.
Author's response: The sentence “This approach can, therefore, transform agroindustrial waste into a valuable biotechnological resource, reducing the environmental impact and supporting the circular economy” (lines 81-83) was deleted. Therefore, a new paragraph has been included in the introduction section as follows: “Therefore, economic and environmental assessments could yield important insights into the project's feasibility before industrial implementation. Moreover, this becomes even more relevant in Chile, where, unlike in European countries like Spain, where OMSW is increasingly valorized through various strategies, it remains a waste stream. No large-scale valorization processes are currently implemented, and the waste is commonly sent to accumulation tanks without further treatment or recovery” (Lines 83 – 89)
Reviewer #2 comment 3: Line 98 “(LCA) of MnP production by A. discolor, comparing the following production alternatives”. It is necessary to justify the choice of this fungus.
Author's response: The authors have included the justification of the use of A. discolor. The following sentence has been added: “Therefore, this study aims to perform both an economic and LCA of MnP production by A. discolor, a Chilean native WRF selected for its high MnP production capacity” (lines 116 -118).
Reviewer #2 comment 4: Figure 1: Is OMSW fermented in aqueous suspension? This needs to be clarified further, as fermentation of solid substrates can occur in aqueous suspension or with low water content.
Author's response: The information required by the reviewer has been included in the M&M section as follows: “All fermentation alternatives were designed based on an aqueous medium” (line 146-147).
Reviewer #2 comment 5: Lines 553-554: I believe this statement is wrong, as there are studies in the literature that evaluated this residue to produce enzymes. It needs to be checked.
Author's response: The authors agree with the reviewer’s observation. To improve clarity, the original sentence was revised and reformulated as follows: “This is the first study to explore MnP production from OMSW using A. discolor, while also incorporating environmental and economic aspects, often overlooked in studies focused mainly on enzymatic yields” (lines 651 - 653).

Reviewer 3 Report
Comments and Suggestions for Authors
This article presents an economic analysis of the production of MnP from the fermentation of olive mill solid residue (OMSW) and subsequent anaerobic digestion of the fermented waste. Some corrections are necessary to improve the text:
- The title needs to be changed, as it gives the impression that fermentation processes were carried out, but this did not occur. This is an article of economic analysis only. It is also unnecessary to include the country Chile in the title;
- Lines 68-69: What is the conventional destination of OMSW? If there is a destination, even if it is less noble, then it cannot be considered waste;
- Line 98: It is necessary to justify the choice of this fungus;
- Figure 1: Is OMSW fermented in aqueous suspension? This needs to be clarified further, as fermentation of solid substrates can occur in aqueous suspension or with low water content;
- Lines 553-554: I believe this statement is wrong, as there are studies in the literature that evaluated this residue to produce enzymes. It needs to be checked.
Author Response
Reviewer #3 comment 1: It is suggested that adding some descriptions or calculations about the carbon dioxide (equivalent) emission reduction of the whole process under different situations.
Author's response: Accordingly, a detailed comparison of COâ‚‚ equivalent emissions among the evaluated alternatives has been incorporated into the manuscript in the following sentences: “According to Table 5, GW and TA impacts are higher in Alternative 2 than in Alternative 1, with values of 879 versus 351.3 kg COâ‚‚ eq for GW and 2.5 versus 1.3 kg SOâ‚‚ eq for TA. This increase is mainly due to the lower MnP yield obtained when using OMSW as a substrate, which requires processing larger volumes of culture medium and greater energy input for sterilisation. Although MnP yields in Alternative 3 are equal to those in Alternative 2, GW and TA impacts are reduced by 69 and 64%, respectively. These reductions are attributed to the heat generated from biogas in the AD stage (i.e., 2,033 MJ/kg MnP produced). As a result, carbon emissions decrease from 879 to 271.5 Kg CO2 eq/Kg MnP, and sulphur dioxide emissions from 2.5 to 0.9 kg SOâ‚‚ eq/Kg MnP.” (Lines 509 - 524).
Reviewer #3 comment 2: It is suggested that describe the replicates and method of statistical analysis in more details so that the contents will be much better.
Author's response: The fermentation assays were conducted in triplicate, and the corresponding statistical analysis is detailed in a previous publication by the authors (Araneda et al., 2023), which is cited in the manuscript. Regarding the biochemical methane potential (BMP) tests presented in this study, these were also performed in triplicate. This information is included in the manuscript as follows: “All assays were performed in triplicate” (Lines 233–234).
